# Microbial diversification is maintained in an experimentally evolved synthetic community

Zahraa Al-Tameemi,[1] Alejandra Rodríguez-Verdugo[1]

**ABSTRACT** Microbial communities are incredibly diverse. Yet, the eco-evolutionary processes originating and maintaining this diversity remain understudied. Here, we investigate the patterns of diversification for *Pseudomonas putida* evolving in isolation and with *Acinetobacter johnsonii* leaking resources used by *P. putida*. We experimentally evolved four experimental replicates in monoculture and co-culture for 200 generations. We observed that *P. putida* diversified into two distinct morphotypes that differed from their ancestor by single-point mutations. One of the most prominent mutations hit the *fleQ* gene encoding the master regulator of flagella and biofilm formation. We experimentally confirmed that *fleQ* mutants were unable to swim and formed less biofilm than their ancestor, but they also produced higher yields. Interestingly, the *fleQ* genotype and other mutations swept to fixation in monocultures but not in co-cultures. In co-cultures, the two lineages stably coexisted for approximately 150 generations. We hypothesized that *A. johnsonii* modulates the coexistence of the two lineages through frequency-dependent selection. However, invasion experiments with two genotypes in monoculture and co-culture did not support this hypothesis. Finally, we conducted an evolutionary "replay" experiment to assess whether the presence or absence of *A. johnsonii* influenced the coexistence of morphotypes at the population level. Interestingly, *A. johnsonii* had a stabilizing effect on the co-culture. Overall, our study suggests that interspecies interactions play an important role in shaping patterns of diversification in microbial communities.

**IMPORTANCE** In nature, bacteria live in microbial communities and interact with other species, for example, through the exchange of resources leaked into the external environment (i.e., cross-feeding interactions). The role that these cross-feeding interactions play in shaping patterns of diversification remains understudied. Using a simple bacterial system in which one species cross-feeds resources to a second species (commensal species), we showed that the commensal species diversified into two subpopulations that persisted only when the cross-feeder partner was present. We further observed loss-of-function mutations in flagellar genes that were fixed in monocultures but not in co-cultures. Our findings suggest that cross-feeding species influence patterns of diversification of other species. Given that nutrient leakage is pervasive in microbial communities, the findings from this study have the potential to extend beyond our specific bacterial system. Importantly, our study has contributed to answering the larger question of whether species evolved differently in isolation versus when interacting with other species.

**KEYWORDS** interspecies interactions, cross-feeding, evolutionary tradeoffs, maintenance of diversity, experimental evolution, synthetic communities

Address correspondence to Alejandra Rodríguez-Verdugo, alejanr1@uci.edu.

The authors declare no conflict of interest.

See the funding table on p. 15.

Long-term experimental evolution studies have demonstrated that microbial populations can undergo significant adaptation and diversification over thousands

of generations (1). Such adaptation involves the acquisition of new genetic mutations that enhance survival and reproduction in specific environments. Often, these adaptive mutations sweep in the population, resulting in a loss of genetic polymorphism (2, 3). However, polymorphisms are sometimes maintained, leading to the formation of new lineages. For example, in the bacterium *Escherichia coli*, acetate specialists evolved from a clonal ancestor that uses glucose and releases acetate as a waste product, resulting in a polymorphic population (4, 5). In addition to cross-feeding, spatial structure can drive microbial diversification by creating distinct ecological niches, which leads to more genetic polymorphism within populations (6–9). For example, during an experiment in static culture conditions, *Pseudomonas fluorescence* diversified into three morphotypes (7). Spatial structure provided an ecological opportunity to types that specialized to the different microenvironments in the flask. Subsequent studies with other species, such as *Burkholderia* and *Bacillus*, have shown that spontaneous diversification due to spatial structure is common (10, 11). These studies have also shown that in order to maintain polymorphisms, fitness tradeoffs are necessary (12). In addition to fitness tradeoffs, negative frequency-dependent selection, in which a rare mutant rises in frequency in a resident population, but loses its fitness advantage when it becomes common, ensures stable coexistence between genotypes (13).

In these previous examples, diversification occurs within species, and intraspecific interactions combined with tradeoffs support polymorphisms (4, 5, 7, 10, 11). Fewer studies have investigated whether other species influence patterns of diversification within species (14). Zhang et al. (14) showed that the diversification in *P. fluorescence* was not affected by the presence or absence of the competitor species *P. putida*. However, the question of how the presence or absence of a species impacts diversification remains little studied. On the one hand, the presence of other species may limit diversification by reducing the number of niches available for diversification (15). On the other hand, other species may promote diversification, based on the observations that interspecific interactions promote coexistence within species (16–18). Thus, after polymorphisms have occurred, another species could mediate the coexistence of newly formed lineages. More generally, species interactions alter effective population sizes, which can influence evolutionary processes (19). For example, competitive interactions, which reduce species' population sizes, have been shown to constrain evolution (20). Less is known about what patterns are expected when species interact positively, for example, when the other species increase their population size. For example, during cross-feeding, microbes exchange metabolites that benefit each other (21). How are patterns of diversification expected to differ when a focal species evolves in isolation compared to when it evolves with another species that cross-feeds resources to it remains an open question.

To address this question, we have used a synthetic microbial community composed of *Acinetobacter johnsonii* C6 and *Pseudomonas putida* KT2440. These bacterial species have been isolated from a creosote-polluted aquifer and have naturally evolved to biodegrade aromatic compounds that make up a significant portion of environmental pollutants (22, 23). In addition to its importance for bioremediation, this synthetic community is considered a model system for studying metabolic and commensal interactions (24–26). The commensalism is based on a unidirectional cross-feeding interaction between *A. johnsonii* and *P. putida*. *A. johnsonii* consumes benzyl alcohol and produces benzoate— an intermediate by-product of benzyl alcohol oxidation—which leaks into the external environment where *P. putida* uses it as a source of carbon and energy (27).

In this study, we build on a previous evolution experiment in which *P. putida* evolved independently and in conjunction with *A. johnsonii* over 200 generations (28). Over the course of this experiment, we observed that populations of *P. putida* diversified into two morphotypes, but this diversification was maintained exclusively in the presence of *A. johnsonii*. To characterize this diversification, we used both genotypic and phenotypic data to address the following questions: How does trait evolution differ in the presence and absence of *A. johnsonii*? What types of mutations underpin the observed pheno-

typic diversity? How do these mutations affect physiology and tradeoffs? Finally, what potential processes underlie the persistence of this diversity in the co-culture conditions?

## MATERIALS AND METHODS

### Growth conditions

All strains were cultured in 10 mL FAB minimal medium [1 mM $MgCl_2$, 0.1 mM $CaCl_2$, 0.003 mM $FeCl_3$, 15 mM $(NH_4)_2SO_4$, 33 mM $Na_2HPO_4$, 22 mM $KH_2PO_4$, and 50 mM NaCl] with either 0.6 mM of benzoate or 0.6 mM benzyl alcohol as the sole carbon source. The medium was run through a 0.2-µm filter and stored in glass treated to remove assimilated organic carbon (AOC). AOC-free glassware was prepared following the protocol by Rodríguez-Verdugo et al. (27). Before running experiments, all strains were acclimated by inoculating 10 µL of frozen glycerol stocks in FAB minimal medium supplemented by either 0.6 mM benzoate or 0.6 mM benzyl alcohol and incubated for 24 h at 26°C with constant shaking (150 rpm).

### Evolution experiment

The evolution experiment had been previously described in Rodríguez-Verdugo and Ackermann (28). Briefly, the experiment consisted of evolving four replicates of *P. putida* in monoculture, four replicates of *A. johnsonii* in monoculture, and four replicates of *P. putida* with *A. johnsonii* (i.e., co-cultures) for 30 days, which equates to approximately 200 generations (~6.6 generations per day). All cultures were grown in batch culture with FAB minimal media and supplemented with 0.6 mM benzyl alcohol. The experimental lines were established from four ancestors of *P. putida* and *A. johnsonii*, isolated from single clones (Fig. S1). Thus, the ancestors diverged by a few generations when the evolution experiment was started. During the evolution experiment, the population densities for the monoculture and the co-culture from replicate #2 were estimated at the end of each day by plating cultures in LB agar with streptomycin (64 µg/mL) and in LB agar with gentamycin (10 µg/mL) to track *A. johnsonii* and *P. putida*, respectively (28).

### Growth curves and resource use traits

To characterize the phenotypic diversification of *P. putida* in monoculture and co-culture, we performed growth curve experiments on 30 single colonies isolated from each of the eight experimental lines and from four ancestral populations. Briefly, bacteria were grown in FAB medium supplemented with 0.6 mM benzoate for 24 h and were then plated on Lysogeny Broth (LB) agar and incubated for another 24 h at 26°C. We randomly selected 30 colonies off the plate and resuspended them into wells filled with 200 µL 1% $MgSO_4$ in a 96-well plate. We then transferred 2 µL of the diluted cells and transferred them to another 96-well plate containing 198 µL of FAB supplemented with 0.6 mM benzoate. The plate was sealed with vacuum grease to reduce evaporation. The plate was incubated in a photospectrometer plate reader (Epoch2, Agilent) at 30°C and continuous linear shaking. Optical density (OD) measurements were taken every 10 min for 24 h. From these growth curves, we estimated four quantitative traits involved in resource use: the maximum growth rate in exponential phase ($\mu_{max}$), the yield ($Y$) defined as the amount of biomass produced per unit of resource, the maximum uptake rate ($V$) defined as the maximum rate of resource consumption per hour and the half-saturation constant ($K$) defined as the resource concentration supporting half-maximum uptake rate. Parameters were estimated from growth curves using a multistep approach (27). First, we estimated the yield (in OD/mM) using the following formula: $Y = \dfrac{OD_{max} - OD_{min}}{R_0}$, with $OD_{max}$ being the maximum OD achieved, $OD_{min}$ the minimum OD at the start of the experiment (both in unitless OD), and $R_0$ the initial concentration of benzoate (=0.6 mM). Second, we modeled the uptake rate (in

mM/OD/h) using the formula: $q = \frac{\mu}{Y}$ where $\mu$ is the instantaneous growth rate ($h^{-1}$) calculated as $\mu = \Delta\ln(OD)/\Delta t$. Third, we modeled the change in benzoate over time using the formula $\frac{dR}{dt} = -\frac{1}{Y}\frac{dP}{dt}$, in which $P$ is the bacterial density (in unitless OD). Finally, we estimated $V$ (in mM/OD/h) and $K$ (in mM) from the function $d(R) = \frac{VR}{K+R}$ fitted to the relationship between the uptake rate and the concentration of benzoate. Parameters were fitted to three growth curves (technical replicates) using Matlab (version R2017a, MathWorks). To quantify the evolution of phenotypic traits, we compared the trait distributions from ancestral populations ($n = 30$) to those from evolved populations ($n = 30$) using a two-tailed $t$ test (R, version 4.3).

## Genome resequencing and variant calling

We extracted DNA from the ancestors' clones, monoculture and co-culture populations, and single clones from large and small morphotypes of all four experimental lines. Single clones were revived from glycerol stocks in 3 mL of LB broth and grown overnight at 26°C with constant shaking (150 rpm). The overnight cultures from monoculture and co-cultures whole populations were grown FAB with 0.6 mM benzoate supplemented with gentamycin (10 µg/mL). We isolated and purified genomic DNA from 1 mL of the overnight cultures using the Wizard DNA purification kit (Promega). Short-read sequencing (2 × 150 bp paired-end reads) was conducted on a NovaSeq6000 S4 flow cell (UCI Genomics High-Throughput Facility). These sequences can be accessed at the NCBI Sequence Read Archive under Bioproject ID number PRJNA623337.

All sequences were aligned to the reference genome of *P. putida* KT2440 (reference NC_002947) using the computational pipeline *breseq* (version 0.36.1) and utility program GDtools (29). We used consensus mode and polymorphism mode to call variants for clones and whole populations, respectively. Each sample was sequenced to a depth of ~600× coverage. Single-nucleotide polymorphisms in genic regions at or above 5% and passing the Fisher's exact test for biased strand distribution were considered. *De novo* non-synonymous mutations were defined as uniquely observed in the evolved sample and not the ancestral clone.

Selection and enrichment statistical tests were implemented using the method described in Rodríguez-Verdugo and Ackermann (28). Briefly, the strength of parallelism at the gene level was estimated using a Poisson distribution: $\left(X = x\right) = \frac{e^{-\lambda}\lambda^x}{x!}$ with $\lambda = \frac{\text{total observed mutations}}{\text{genome size in bp}} \times \text{size of the target gene in bp}$. The $P$ value was estimated from the Poisson cumulative expectation, $P(x \geq \text{observed}, 1)$. The cellular functions under selection were determined using the statistical overrepresentation test from Panther (pantherdb.org), version 19.0.

## Physiological assays

### Biofilm formation

We measured biofilm formation in the *P. putida* ancestors and the evolved morphotypes using the method described in Martínez-García et al. (30). Samples were inoculated into FAB media supplemented with 0.6 mM benzoate and incubated for 24 h at 26°C with constant shaking (150 rpm). The tubes were then vortexed to homogenize the culture and the $OD_{600}$ was adjusted to 0.5. We added 2 µL of the adjusted acclimated culture to a 96-well plate containing 198 µL of FAB and 0.6 mM benzoate and was incubated at 26°C with no shaking. Each sample was inoculated into five wells that serve as technical replicates. After 24 h, the plate was removed from the incubator, and the $OD_{600}$ was measured using the plate reader. The media with planktonic cells were discarded by gently tapping the plate and were washed three times with water. The cells attached to the plate were stained by adding 200 µL of 0.1% (wt/vol) crystal violet into each well and were incubated on the bench top for 30 min. We then disposed of the crystal violet and

gently washed the plate two to three times with deionized water. The plate was left to completely dry. We added 200 µL of 33% (wt/vol) acetic acid to each well, pipetting up and down until it was completely homogenized. The $OD_{595}$ was measured, and an index of biofilm formation was calculated, which was the crystal violet stained biofilm ($OD_{595}$) divided by the initial culture with planktonic cells ($OD_{600}$). The higher the biofilm index value, the greater the biofilm formation. These experiments were replicated three times. We conducted a two-tailed $t$ test to determine if there were significant differences in biofilm formation in the evolved morphotypes ($n = 3$) compared to the ancestor ($n = 3$).

### *Motility*

We performed a swimming assay on the ancestors and evolved morphotypes of *P. putida* using 0.3% (wt/vol) soft agar (30, 31). We inoculated 2 µL of an acclimated culture onto FAB soft agar supplemented with 0.6 mM benzoate and incubated the plates at 30°C. We measured the halo around the initial inoculum after 24 h to determine whether the evolved morphotypes' swimming abilities differed from the ancestors' swimming abilities. A larger halo indicates higher motility, while the absence of a halo suggests a lack of flagellar motility.

### Reciprocal invasion assays

To test if frequency-dependent selection is a possible process for maintaining emergent diversity, we performed reciprocal invasion experiments. These experiments consisted of culturing two morphotypes (large and small) derived from one of the evolved replicates (replicate #2) at a frequency of 5%, 10%, 50%, 90%, and 95% to see if they can invade when rare in the presence and absence of *A. johnsonii*.

We used a large and a small clone of *P. putida* (referred as large and small genotypes) from the co-culture #2 isolated at generation 200, which have been genetically characterized in this study (see "Genome resequencing and variant calling"). For *A. johnsonii*, we used a single clone from co-culture #2 isolated at generation 200, which was previously sequenced (28). We revived these clones from glycerol stocks by inoculating 10 µL of frozen culture in FAB medium supplemented with 0.6 mM benzoate (for *P. putida*) and with 0.6 mM benzyl alcohol (for *A. johnsonii*) and incubating them at 26°C with 150 rpm for 24 h. The OD was then adjusted to 1.0 for both genotypes to standardize the starting ratio. We tested five initial frequencies with a given genotype starting from rare (5% and 10%), equal (50%), or common (90% and 95%). These frequencies were achieved by adjusting the volumetric ratios. For example, for a starting ratio of 10%, 10 µL of the large genotype, and 90 µL of the small genotype were inoculated into 9.9 mL of fresh media. For the condition without *A. johnsonii* (i.e., monoculture), the medium was supplemented with 0.6 mM benzyl alcohol. Co-cultures and monocultures were grown at 26°C with constant shaking (150 rpm) for 24 h. We counted colony-forming units (CFU) at the start of the experiment (T0) and the end of the growth cycle by plating cultures onto LB agar plates supplemented with gentamicin (10 µg mL$^{-1}$). We included three technical replicates per experiment.

The Malthusian parameters for the two genotypes were calculated from cell densities at the initial and final time points, $m = \ln (N_f/N_0)$, with $N_0$ and $N_f$ being the initial and final density after 24 h, respectively. From the Malthusian parameters, we estimated the selection rate constant $r_{ij}$ corresponding to the relative growth rate of a morphotype against its competitor. We used the formula $r_{ij} = \dfrac{m_i - m_j}{1 \text{ day}}$ for a pair of competitors $i$ and $j$ (32, 33). If the value of $r_{ij}$ was significantly larger than 0, the genotype $i$ could invade the genotype $j$. If both the large and small genotypes had a positive $r$ value when started from rare, the pair was considered to stably coexist through frequency-dependent selection.

## Evolutionary "replay" experiment

To better understand the effect of *A. johnsonii* on the maintenance of *P. putida*'s diversity in co-culture, we revived and evolved one of the co-cultures (replicate #2) to see if diversity loss occurred when *A. johnsonii* was removed. We revived co-culture #2 from frozen glycerol stocks stored at −80°C on the 21st transfer day; ~138 generations of evolution. We inoculated 100 µL of the frozen stock into 9.9 mL of FAB with 0.06 mM of benzyl alcohol. To remove *A. johnsonii* from the community, we added 60 µg/mL of gentamycin to the media and incubated the culture at 26°C and 150 rpm for 1 day. The removal of *A. johnsonii* was visually confirmed through plating the culture on LB agar plates with streptomycin (64 µg/mL). Two cultures—one *A. johnsonii*-free community and one "intact" community (control co-culture)—were used to inoculate six culture tubes (technical replicates) with 9.9 mL FAB minimal medium supplemented by 0.6 mM benzyl alcohol. For each daily passage, we transferred 100 µL of culture into 9.9 mL of fresh medium, following the same methodology used for the evolution experiment (28). We did serial transfers for eight days to replay ~50 generations of evolution (simulating the last phase of the 200 generations evolution experiment). At the end of a 24-h growth cycle, we plated 100 µL of culture in LB agar plates supplemented with antibiotics to obtain colony counts.

## RESULTS

### Phenotypic diversification persisted only in co-culture

After 200 generations of evolution, we observed diversification of *P. putida* in terms of colony morphology relative to the ancestor (Fig. 1A). All four replicates were bimodal for colony size and consisted of one cluster of strains that formed larger colonies on LB agar plates compared to the ancestor (referred as "large morphotype"), and one cluster that formed colonies the size of the ancestor (referred as "small morphotype"). This

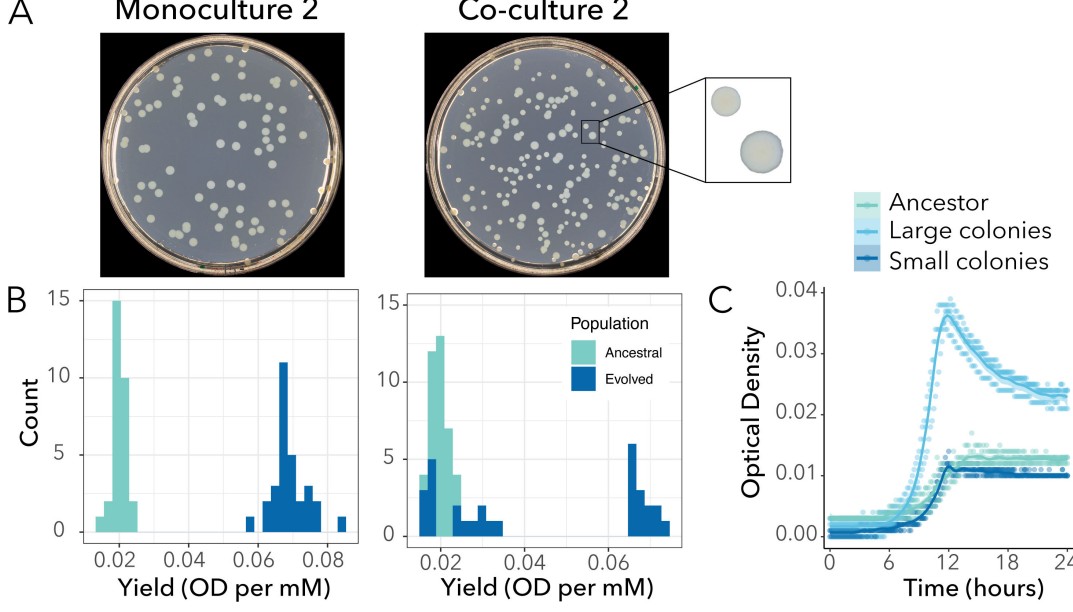

**FIG 1** *P. putida* population evolved in monoculture is monomorphic, while the population evolved in co-culture diversified into two subpopulations after 200 generations. (A) Agar plates illustrating the difference in colony morphologies between the evolved population in monoculture (left panel) and the evolved population in co-culture (right panel) for replicate #2. (B) Histograms showing the differences in yield distributions between the ancestral and the populations from replicate #2 evolved in monoculture (left panel) and co-culture (right panel). In monoculture, *P. putida*'s population evolved higher yield (*t* test, $P < 2.2 * 10^{-16}$). In co-culture, *P. putida*'s population evolved bimodality (Hartigans' dip test for multimodality, $P < 2.2 * 10^{-16}$). The results from the other replicates and parameters are available in Fig. S2 and S3. (C) The optical density ($OD_{600}$) over 24 h is plotted for three randomly selected colonies (i.e., three replicates) sampled from the ancestral population, the subpopulation forming small morphotypes and the subpopulation forming large morphotypes from co-culture #2.

diversification in colony size was only observed when *P. putida* evolved in co-culture with *A. johnsonii*. In monocultures, populations only formed larger colonies than the ancestor.

Next, we quantitively characterized this diversification in four traits involved in resource use. These traits are interdependent—that is, they all are derived from growth curves—but they provide insights into subtle differences at different phases of the growth cycle. In all monoculture and co-culture replicates, the phenotypic variance for each trait changed compared to the variance of the ancestral population (Fig. S2 and S3). Changes in yield were the most pronounced. We found that *P. putida* grown in co-culture with *A. johnsonii* had a different distribution of yields compared to monoculture (Fig. S2). For example, co-cultures of replicate #2 showed a bimodal distribution, with one subpopulation having a similar yield to the ancestor and the other having a higher yield to the ancestor (Fig. 1B, right panel). On the other hand, the monoculture had a distribution shifted toward higher yield (Fig. 1B, left panel). We confirmed that the large morphotypes previously observed on LB agar plates had higher yields compared to their ancestors, and the small morphotypes had similar yields to their ancestors. For example, large morphotypes from replicate #2 had, on average, a 3.4-fold higher yield than the ancestor ($0.068 \pm 0.001$ and $0.020 \pm 0.001$ OD per mM for the large and ancestral morphotypes, respectively; *t* test, $P < 2.2 * 10^{-16}$). The small morphotypes achieved a yield of $0.023 \pm 0.002$ OD per mM, which is comparable to that of the ancestor (*t* test, $P = 0.122$).

Finally, leveraging the fact that large morphotypes with higher yield formed larger colonies on LB agar plates that could be distinguished from the ancestral morphotype, we tracked the evolution of phenotypic diversification for one of the replicates (replicate #2) during our experiment. After approximately 40 and 70 generations of evolution, we observed the emergence of a large morphotype in monoculture and co-culture, respectively (Fig. 2). In co-culture, the large and small morphotypes coexisted for the remaining 130 generations. Instead, in monoculture, the large morphotype rapidly outcompeted the ancestral morphotype, driving it to extinction after 140 generations. Thus, diversification only persisted in co-culture.

## Morphological diversification in *P. putida* is based on genetic diversification

To determine if the observed phenotypic diversity in the evolved lines (Fig. 1) reflected specific mutations, we performed whole population sequencing to identify *de novo* mutations. Combined, all eight evolved lines accumulated 29 point mutations in open reading frames (Table S1). Most of these mutations were non-synonymous

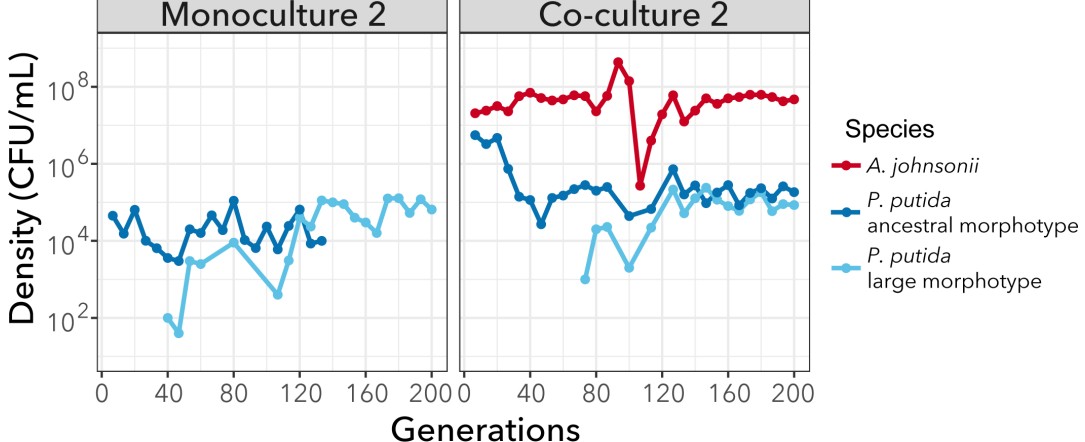

**FIG 2** Diversification was maintained in co-culture but not in monoculture. The large morphotype appears in the monoculture after approximately 50 generations and outcompetes the ancestral morphotype. In co-culture, both morphotypes coexist until the end of the experiment. Population density (CFU mL$^{-1}$) trajectories from replicate #2 are shown over 200 generations. *A. johnsonii*'s densities are from reference 28. Each point represents the final density of each species and morphotypes after daily cycles of growth; lines connecting circles help visualization and do not have any biological meaning.

(non-synonymous/synonymous ratio = 8.7), indicating a strong signal of adaptive evolution. In addition to point mutations, some lines accumulated short insertions and deletions (<30 bp) and one line had a deletion of 130 bp (Table S1). Mutations targeted functions related to positive regulation of cell motility (GO:2000147), negative regulation of extracellular matrix assembly (GO:1901202), positive regulation of cell-substrate adhesion (GO:0010811), and DNA-template transcription (GO:0006351). There were no differences in total number of mutations in monocultures compared to co-cultures (Fig. 3A).

Next, we looked at patterns of convergent evolution. We observed parallel evolution at different levels. The exact same non-synonymous mutation in the *cmoB* gene was shared among three experimental lines (Table S1). Other mutations also targeted the *cmoB* gene, suggesting it is under strong positive selection. The CmoB enzyme is involved in post-transcriptional modifications of tRNA molecules, but its functional relevance remains underexplored. Another gene repeatedly targeted was the *fleQ* gene, encoding FleQ, a master transcriptional regulator of flagella and biofilm formation. Four out of eight lines had mutations in the *fleQ* gene. This parallelism suggests this gene is under strong selection based on the expectation that observing four random mutations in a 1476 bp gene is extremely low (Poisson cumulative expectation, $P = 2 * 10^{-12}$).

More generally, approximately one-third of mutations (including deletions and insertions) targeted the same genes in monocultures and co-cultures, while the rest were either found exclusively in monocultures (~43%) or in co-cultures (~25%; Fig. 3B). For example, mutations in the *uvrY* gene encoding a UvrY/SirA/GacA family response

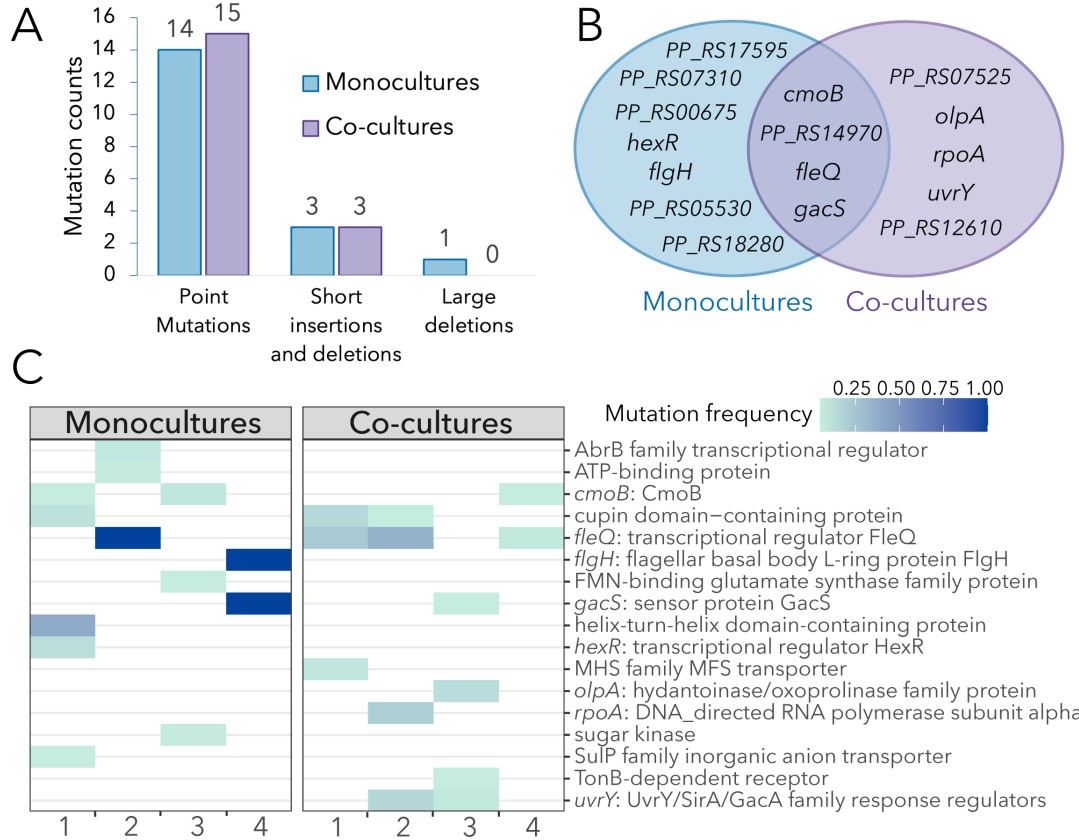

FIG 3 Mutational landscape of monocultures compared to co-cultures. (A) Mutations count according to types: point mutations (both synonymous and non-synonymous mutations), short insertions or deletions (i.e., short in-frame indels), and large deletions (>30 bp). (B) Venn diagram showing convergence of mutations in genes. (C) Heat map showing the frequency of non-synonymous mutations (genes = rows) in four monocultures and four co-cultures (columns). Genes with mutations at frequencies above 0.05 are shown. The frequency of each mutation is shown by different intensities of blue: darker blue corresponds to mutations at higher frequencies.

regulator were observed exclusively in two co-cultures and may be indicative of adaptations to the biotic conditions (presence of *A. johnsonii*). Interestingly, hard sweeps were only observed in monocultures; that is, two lines evolved in monocultures had mutation fixed in the population (i.e., at a frequency of 100%; Fig. 3C). One of the monocultures (from replicate #2) had a frameshift mutation in the *fleQ* gene fixed in the population (Table S1). The other monoculture (from replicate #4) had two fixed mutations: one frameshift mutation in the *flgH* gene coding for flagellar basal body L-ring protein FlgH and one non-synonymous mutation in the *gacS* gene coding the sensor protein GacS. Mutations in the *fleQ* and *gacS* genes were also observed in co-cultures but were not fixed in the population. For example, mutations in the *fleQ* gene were observed at a frequency of 23.1%, 35.6%, and 7.5% in replicates #1, #2, and #4, respectively.

Finally, we investigated the linkage of these mutations within single clones that formed large morphotypes (from monoculture and co-culture conditions) and small morphotypes (from co-cultures). On average, each clone had one mutation, although some clones had two or no mutations (Table 1). Three out of four small morphotypes had one non-synonymous mutation. Thus, despite having a similar yield to the ancestor, they differed from it by one mutation. Most of these mutations were present at low frequency in the population (<5%) and affected unannotated genes. One exception was the small morphotype from replicate #3, which had a 1 bp deletion in the *oplA* gene and was present at a 12.9% frequency in the population (Table S1). The *oplA* gene codes for 5-oxoprolinase A, which has been associated with the catabolism of amino acid derivatives (34, 35) and surface attachment abilities (36). Instead, the large morphotypes were often associated with high-frequency mutations in the *fleQ*, *flgH*, and *gacS* genes. Notably, two of these large morphotypes were single mutants; that is, they had only one *fleQ* mutation in the ancestral background (Table 1). Thus, their higher yield compared to the ancestor could be directly attributed to the effect of the mutation. That said, two of the large morphotypes had no mutations in coding regions but had mutations in intergenic regions. It is possible that these mutations are associated with promoters, but the genetic bases for their higher yield require further investigation.

Taken together, we identified the *fleQ*, *flgH*, and *gacS* mutations as putative adaptive based on the following: (i) they were present at high frequency in the populations (e.g., some were fixed), (ii) they displayed high level of parallelism, notably *fleQ*, and (iii) single mutants had higher yield compared to the ancestor.

**TABLE 1** Frequency of *de novo* mutations in single clones of *P. putida* in monoculture and co-culture after 200 generations[a]

| Clone | Morph. | Position | Mutation | Gene | Freq. in pop. | Description |
|---|---|---|---|---|---|---|
| Mono. 1 | Large | 3,987,060 | Δ1,094 bp | *oplB*/*oplA* | <5% | –[c] |
| Mono. 2 | Large | 4,964,741 | 2 bp→ C[b] | *fleQ* | 100% | Transcriptional regulator FleQ |
| | | 5,715,528 | N65N | *tatB* | 100% | TatB |
| | | 6,029,059 | A134A | *kauB* | 100% | KauB |
| Mono. 3 | Large | –[c] | – | – | – | – |
| Mono. 4 | Large | 1,843,830 | D322H | *gacS* | 100% | Sensor protein GacS |
| | | 4,978,570:1 | +C | *flgH* | 100% | Flagellar L-ring protein |
| Co-culture 1 | Large | 4,964,681 | A226V | *fleQ* | 23.1% | Transcriptional regulator FleQ |
| Co-culture 1 | Small | – | – | – | – | – |
| Co-culture 2 | Large | 4,964,300 | L353Q | *fleQ* | 35.6% | Transcriptional regulator FleQ |
| Co-culture 2 | Small | 3,990,098 | V128G | PP_RS18280 | <5% | Helix-turn-helix domain-containing protein |
| Co-culture 3 | Large | 4,635,453 | Y101* | *uvrY* | <5% | UvrY/SirA/GacA family response regulator transcription factor |
| Co-culture 3 | Small | 3,987,689 | Δ1 bp | *olpA* | 12.9% | Hydantoinase A/oxoprolinase family protein |
| Co-culture 4 | Large | – | – | – | – | – |
| Co-culture 4 | Small | 4,025,185 | A172T | PP_RS18445 | <5% | HlyD family efflux transporter periplasmic adaptor subunit |

[a]Monoculture data are extracted from our previous study (28); co-culture data were generated in this study.
[b]Deletion of first base and substitution of second base with C.
[c]– indicates that no mutations were found.

## Putative adaptive mutations are associated with loss of motility and biofilm formation

In the previous section, we identified putative adaptive mutations in flagellum genes (two of them frameshift mutations). Previous studies have suggested that loss-of-function mutations in flagellar genes are adaptations to the shaking culture conditions. This is because inactivating flagellar motility (which is not necessary in well-mixed environments) saves energy that can be allocated to other cellular functions, for example, building biomass (30). To test this hypothesis, we explored the mutants' abilities to swim and form biofilms. The motility assay showed a complete lack of motility in the *fleQ* mutants compared to their ancestors (Fig. 4). The large morphotype with mutations in *flgH* and *gacS* also had a reduction in swimming. The other large morphotypes with mutations in genes other than flagellum genes retained the ability to swim. This suggests that their advantages are related to other traits that do not involve flagellar motility. Conversely, a clone from co-culture #4, which formed a small morphotype, lacked motility. This small morphotype had a non-synonymous mutation in the gene PP_RS18445 encoding the HlyD family efflux transporter periplasmic adaptor subunit (Table 1). Therefore, mutations in genes other than flagellum genes can affect motility as well.

Mutations in flagellar genes were also associated with a significant reduction in biofilm formation (Fig. 5). The *fleQ* mutants isolated from co-cultures #1 and #2, and monoculture #2, exhibited significant reductions in biofilm formation compared to their ancestors (*t* test, *P* values of 0.003, 0.023, and 0.008, respectively). The *flgH* mutation (large morphotype from monoculture #4) also had reduced biofilm formation, but the difference was marginally significant (*t* test, $P = 0.068$) (Fig. 5).

In conclusion, these results support the hypothesis that the fitness advantage of the flagellar mutants resides in knocking down swimming and reducing biofilm formation which may save energy that can be invested in building biomass, as previously suggested.

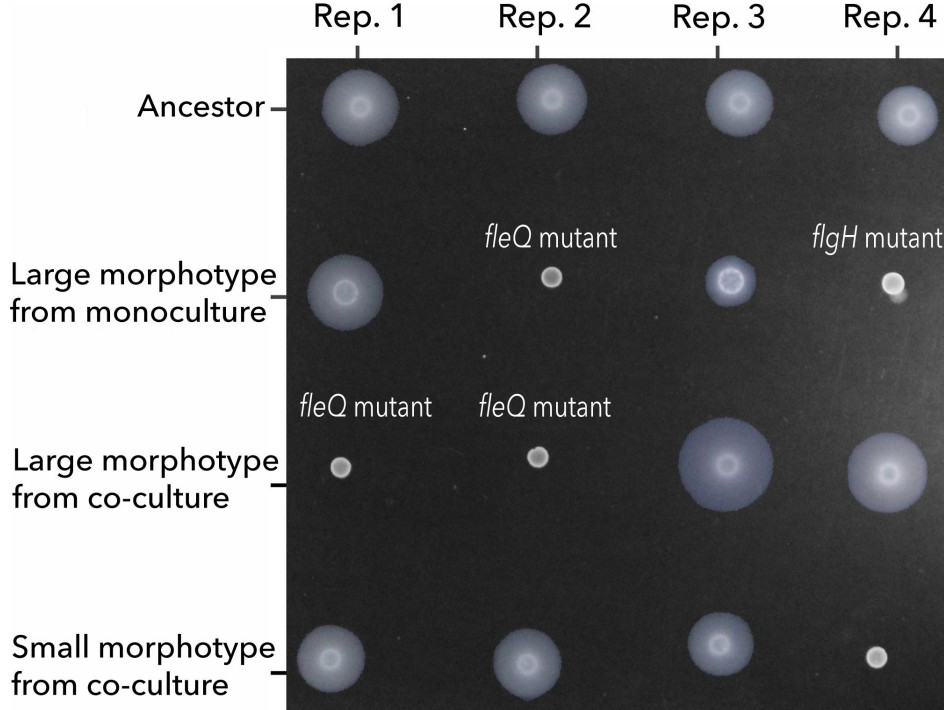

**FIG 4** Swimming assay shows the loss of flagellar motility in the large morphotypes with flagellar mutations. All samples were spotted on 0.3% (wt/vol) FAB soft agar. The absence of a halo around the initial inoculum indicates the loss of flagellum-driven motility.

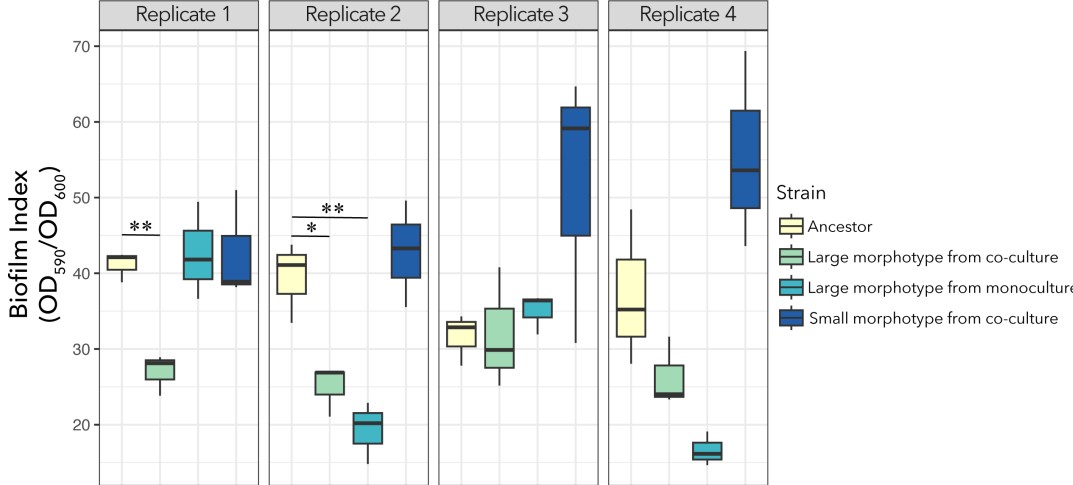

**FIG 5** Biofilm assay shows that *fleQ* mutants produce less biofilm than the ancestor. Biofilm index (CV/OD$_{600}$) after a 24-h incubation period. All *fleQ* mutants (i.e., large morphotypes from co-culture replicate #1, and from the co-culture and monoculture from replicate #2) had a significant reduction in the biofilm index compared to their respective ancestors. Note: * = 0.01 < *P* < 0.05, ** = *P* < 0.01.

## Frequency-dependent selection does not explain the coexistence of genotypes in one of the co-cultures

As previously stated, loss-of-function mutations in flagellar genes are likely to be advantageous and outcompete the ancestor in monoculture conditions. Despite their fitness advantage, they do not sweep to fixation in populations evolved in co-cultures. We wondered if the small genotypes (which are genetically different from the ancestor) have an advantage over the flagellar mutations and whether this advantage is dependent on the presence of *A. johnsonii*. We also wondered if coexistence is maintained by negative frequency-dependent selection. To address these questions we focused on the monoculture and the co-culture populations from replicate #2. In both of these populations, the *fleQ* mutation is present at high frequency but it is only fixed in monoculture (Fig. 3C). We performed a reciprocal invasion experiment between two clones—the *fleQ* genotype and the small genotype—in the presence/absence of *A. johnsonii*. The reciprocal invasion assay did not show significant differences in the rare genotypes' ability to invade depending on whether *A. johnsonii* was present (Fig. S4). The small genotype tended to have a higher fitness relative to the *fleQ* genotype, but these differences were not statistically significant (Fig. 6; Table S2). Finally, we tested whether the relative fitness of the small genotype differed in monoculture than in co-culture by pooling data across frequencies. We did not find significant differences in the small genotype's relative fitness, whether *A. johnsonii* was present or absent (two-tailed *t* test, d.f. 28, *P* value = 0.814) (Fig. 6). In conclusion, these results suggest that frequency-dependent selection is not a process driving coexistence between the *fleQ* and small genotypes from replicate #2, and the presence of *A. johnsonii* does not influence the relative fitness of the small genotype.

## *A. johnsonii* influences patterns of phenotypic diversity in co-culture

In the previous section, we saw that the coexistence of the two genotypes was not explained by the presence of *A. johnsonii*. It is possible that our representative genotypes did not capture the stable coexistence dynamics observed at the population level. Thus, we hypothesize that *A. johnsonii* plays a role in maintaining polymorphism but its effect is only captured considering the entire genetic diversity. To further explore the effect of *A. johnsonii* on the maintenance of phenotypic diversity (as a proxy of genotypic diversity), we did an evolutionary "replay" experiment. That is, we revived the population from co-culture #2 at 138 generations which was already bimorphic (Fig. 2).

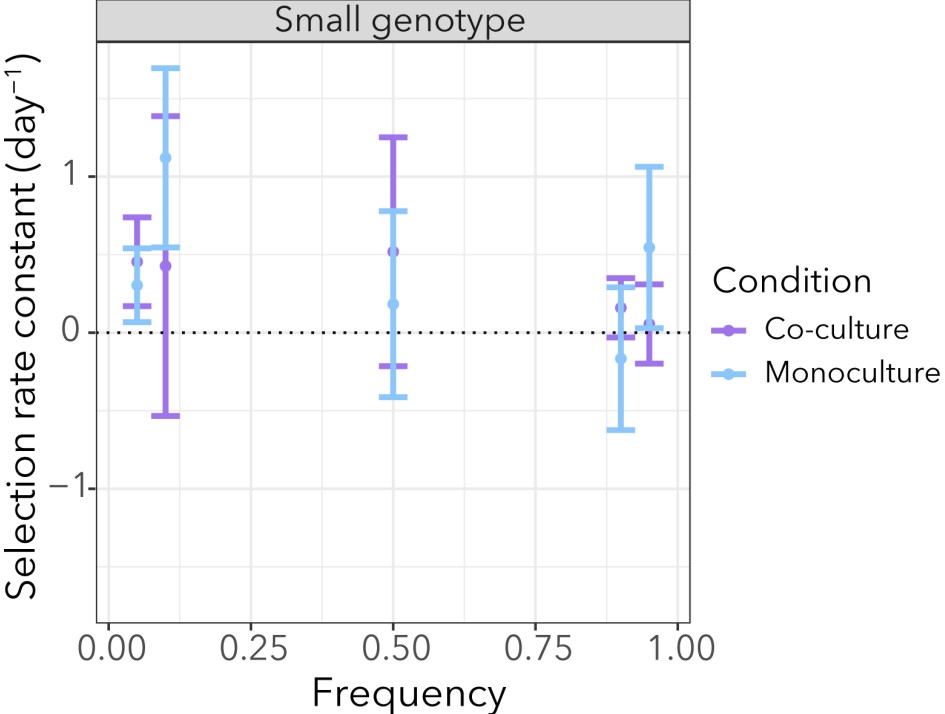

**FIG 6** The relative growth rate of the small genotype was not affected by growth conditions (monoculture or co-culture). Each point shows the average of the selection rate constant and the error bars represent the standard error from six replicates. Despite a tendency for the small genotype to have a fitness advantage over the *fleQ* genotype (i.e., positive selection rate constant), the values were not significantly different from 0 (Table S2).

We then established two populations that we further evolved for ~50 generations with and without *A. johnsonii*. We reasoned that if *A. johnsonii* is important for maintaining diversity, phenotypic diversity would decline when *A. johnsonii* is removed. We observed that, in the condition without *A. johnsonii*, both morphotypes gradually declined but eventually recovered (Fig. 7). Interestingly, the small morphotype became dominant at the end of the experiment and achieved approximately fourfold higher density than

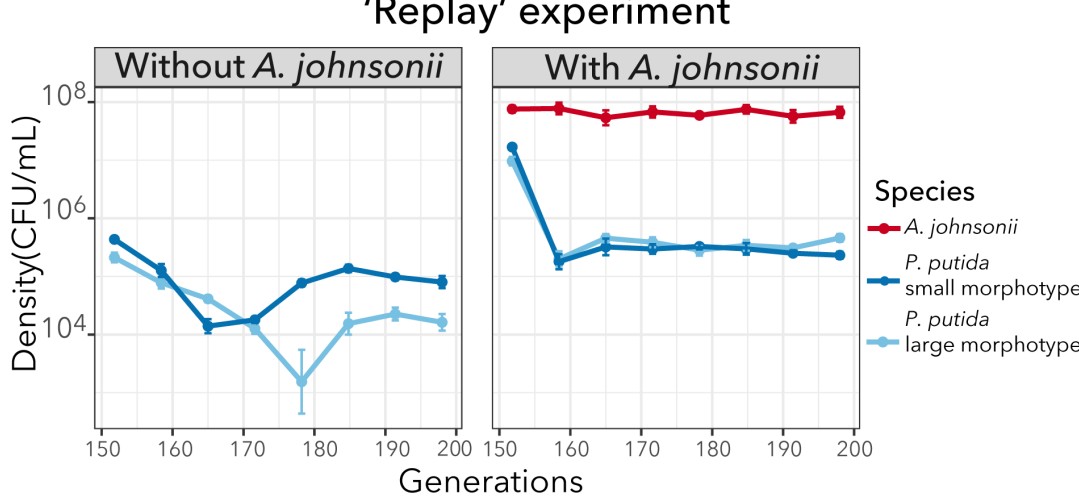

**FIG 7** *P. putida* population dynamics were reproducible when we replayed the evolution experiment from an intermediate time point but were altered when *A. johnsonii* was removed. Mean population densities (CFU mL$^{-1}$) of six replicates with their standard error are represented over 50 generations. The population trajectories for each replicate are shown in Fig S5.

the large morphotype (two-tailed $t$ test, d.f. = 6, $P$ = 0.005). In contrast, population densities were more stable over time in the condition where *A. johnsonii* was maintained (i.e., control "replay" experiment; Fig. S5). In this condition, the large and small morphotypes achieved similar cell densities, but the large morphotype achieved, on average, twofold higher density than the small morphotype at generation 200 (two-tailed $t$ test, d.f. = 6, $P$ = 0.018). However, for most of the experiment, the density differences between the two morphotypes were not significantly different (Fig. 7). Taken together, although we did not see the extinction of one of the morphotypes when *A. johnsonii* was removed, our results suggest that *A. johnsonii* "buffers" population dynamics and prevent the dominance of one morphotype. The mechanisms by which *A. johnsonii* influences population dynamics is an open question that requires further investigation.

## DISCUSSION

The results from this study show that diversity can arise, even in a well-shaken environment with low resources. Phenotypic and genotypic diversity arose in both monoculture and co-culture conditions; however, it persisted only in co-culture conditions. This suggests that the presence of *A. johnsonii* influences evolutionary patterns.

When comparing monocultures with co-cultures, we did not observe differences in the total number of mutations or proportion of non-synonymous to synonymous mutations. This pattern agrees with other systems—for example, *Lactobacillus plantarum* evolving with *Saccharomyces cerevisiae* (20), although a higher rate of adaptation in co-culture conditions has been observed in other studies (37). In our study, an interesting difference between monocultures and co-cultures was that hard sweeps were only observed when *P. putida* evolved in monoculture. Piccardi et al. (37) observed a similar pattern when comparing species evolved in isolation versus in a four-species community (37). They attributed hard sweeps to a sharp decline in population sizes at the beginning of the experiment, which occurred only in monocultures (i.e., drift). Although in our systems, population sizes in monocultures are one order of magnitude lower than in co-cultures (Fig. 2), they are still large (~$10^5$ CFU/mL). Therefore, we do not think that hard sweeps are attributed to drift but rather selection. We showed multiple lines of evidence supporting that mutations targeting the *fleQ*, *flgH*, and *gacS* genes are under strong positive selection. In addition, mutations in these genes have been frequently observed in other evolution experiments using similar culture conditions (minimal media and shaking environments) but with different stresses (38, 39). Thus, it is possible these mutations are general adaptations to culture conditions. Flagellar mutations reducing or eliminating motility have been shown to be advantageous in different culture conditions (40, 41). The consensus explanation for this advantage is that the loss of flagellum motility alleviates a large metabolic burden that can be invested in biomass production (30, 42). All three mutations have been shown to have an effect on flagellar motility. GacS negatively regulates flagellum formation and cell motility based on observations that *gacS* mutants are highly flagellated and have greater swimming abilities (43, 44). In our study, the *gacS* mutation was in the same genetic background as the *flgH* mutation (Table 1). It is possible that the two mutations have opposite effects on motility, which may explain why the *flgH* mutant did not fully lose motility compared to the *fleQ* mutant (Fig. 4). Importantly, regulation of cell motility seems to be an important target of selection which may explain why flagellar mutations outcompeted the ancestor and swept to fixation in monocultures. It is important to mention that other mutations present in the populations at lower frequencies also showed improved fitness (higher yield). One of these mutations was a nonsense mutation in the *uvrY* gene, which is homologous to *gacA*. The *gacA* gene codes the response regulator GacA of the two-component system GacA/GacS and thus may be involved in the regulation of cell motility, as well (45).

A puzzling observation is that these mutations did not sweep to fixation in co-cultures. One possibility is that clonal interference is more pronounced in co-cultures due to the larger population sizes (46). That is, the *fleQ*, *flgH*, *gacS*, and *uvrY* mutations may

still be advantageous in co-culture conditions, but they do not fix in the population because they compete with other adaptive mutations. The question remains: what is the advantage of the other mutations, and what is their functional relevance? One possibility is that their advantage is related to the presence of *A. johnsonii*, which leaks benzoate into the medium. The excretion or leakage of nutritional by-products has been shown to lead to the construction of new niches, which can affect the generation and maintenance of biodiversity (47, 48). Thus, we hypothesized that *A. johnsonii* creates transient patches of resources, and this benefits the clones that retain flagellar motility and biofilm formation abilities (Fig. 4 and 5) because they are better at feeding on the benzoate excreted by *A. johnsonii* (i.e., "foraging" strategy) compared to the clones that lose their ability to swim and form less biofilm. We based this hypothesis on the observation that one of the small morphotypes had a 1 bp deletion in the *olpA* gene coding OlpA, the 5-oxorpolinase subunit A (Table 1). Disruption of OlpA has been shown to increase surface attachment (36). Furthermore, another study showed that motility increases the encounter rate between cross-feeding partners and cell-cell adhesion in a well-mixed environment (49). Nevertheless, we did not find support for this hypothesis when directly competing the small genotype—which retains the ability to swim and form biofilms—against the *fleQ* mutant, which cannot swim. One possibility is that swimming and biofilm formation are not the traits under selection, and the advantage resides in other traits. For example, *A. johnsonii* may have evolved to antagonize *P. putida* and the small genotypes have mutations that make them resist better these selective pressures. Furthermore, it is important to acknowledge that we only focused on two genotypes from one replicate, which is only a subset of the whole diversity of mutations found in co-cultures.

Our study has provided valuable insights into differences in patterns of phenotypic and genotypic diversification in the presence or absence of a distantly related species. However, it is important to acknowledge certain limitations in interpreting the results. First, while 200 generations were enough time to observe diversification, we refrain from making a definitive conclusion about whether the diversity is transient (short-lived) or if it can be sustained long term. Another limitation in interpreting these results is the lack of control for the benzoate excreted by *A. johnsonii*. Although benzoate was not externally supplemented to the *P. putida* monocultures, they relied on residues of organic carbon present in the culture medium, that is, assimilable organic carbon or AOC (28). Thus, an important control to discern if differences in diversification patterns are only due to differences in resource concentration (e.g., impacting population sizes) or can be attributed to other biotic feedbacks, future evolution experiments should add an external source of benzoate to the monocultures. Such experiments would allow us to determine how important it is that benzoate is produced by *A. johnsonii* (i.e., a biogenic resource supply) versus being externally supplied to the medium.

## ACKNOWLEDGMENTS

We thank Jennifer Martiny, Brandon Gaut, and the Rodriguez-Verdugo lab members for providing useful feedback during this research. We also thank two anonymous reviewers for their thoughtful comments that helped us to improve the paper.

This work was supported by the National Science Foundation under Grant no. DEB-2234627. A.R.-V. was supported by UC Irvine's Charlie Dunlop School of Biological Sciences.

## AUTHOR AFFILIATION

[1]Department of Ecology and Evolutionary Biology, University of California, Irvine, Irvine, California, USA

## AUTHOR ORCIDs

Alejandra Rodríguez-Verdugo (iD) http://orcid.org/0000-0002-2048-129X

## FUNDING

| Funder | Grant(s) | Author(s) |
|--------|----------|-----------|
| National Science Foundation (NSF) | DEB-2234627 | Alejandra Rodríguez-Verdugo |

## DATA AVAILABILITY

The genome sequences have been deposited at the National Center for Biotechnology Information (NCBI) Sequence Read Archive (SRA) under BioProject ID number PRJNA623337, accession numbers SRX22321713 to SRX22321732. All the other data generated or analyzed during this study are included in this published article and its supplemental materials.

## ADDITIONAL FILES

The following material is available online.

### Supplemental Material

**Supplemental tables and figures (mSystems01053-24-S0001.pdf).** Fig. S1–S5 and Tables S1 and S2.

### Open Peer Review

**PEER REVIEW HISTORY (review-history.pdf).** An accounting of the reviewer comments and feedback.

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
