## [Reviewer comments · mSystems]

Microbial diversification is maintained in an experimentally evolved synthetic community

Zahraa Al-Tameemi and Alejandra Rodríguez-Verdugo

Corresponding Author(s): Alejandra Rodríguez-Verdugo, University of California Irvine

Review Timeline:

Submission Date:	August 5, 2024
Editorial Decision:	August 27, 2024
Revision Received:	September 10, 2024
Accepted:	September 11, 2024

Editor: William Harcombe

Reviewer(s): The reviewers have opted to remain anonymous.

Transaction Report:

DOI: <https://doi.org/10.1128/msystems.01053-24>

Re: mSystems01053-24 (Microbial diversification is maintained in an experimentally evolved synthetic community)

Dear Dr. Alejandra Rodríguez-Verdugo:

Thank you for your work addressing concerns with the manuscript. Several further minor issues were raised for the authors consideration.

Revision Guidelines

Sincerely,
William Harcombe
Editor
mSystems

Reviewer #1 (Comments for the Author):

The authors have done an excellent job in revising their manuscript. I was quite critical regarding the storyline and conclusions when reading the initial version of the paper. Now, I'm pleased to see that the manuscript is much improved and attractive to read. I appreciate that the authors conducted extra experiments. The data on negative-frequency dependent selection has

become much stronger and while still negative, we can now reject this hypothesis with high confidence. The evolutionary replay experiment is a great gain for the paper. Although the results do not recover the selective sweeps observed in the initial mono-culture replicates, they clearly show the stabilising effect of *A. johnsonii* in co-cultures. I have a few extra comments.

1. The results of the replay experiment could be highlighted more clearly. I propose to mention them in the abstract.
2. Abstract, lines 28-30. The sentence reads odd as it states that the two morphotypes co-exist (with one of them dominating), while line 24 emphasises a selective sweep. This causes confusion. I assume this sentence refers to the replay experiment. I thus recommend to explicitly state this and to emphasise the key results: *A. johnsonii* has a stabilising effect.
3. Line 191. In this section, phenotypic assays are described and not physiological tradeoffs. Please clarify.
4. Fig. 6 and lines 399-401. Since there is no frequency effect, it would be possible to pool the data across frequencies to conduct a global analysis with higher statistical power. I.e., one could test whether the relative fitness of the small genotype is lower in co-culture than in mono-culture, which would point towards a stabilising effect in co-culture.

Reviewer #2 (Comments for the Author):

The authors have addressed my concerns - thank you.

I have only one minor edit to point out - Line 698 "got altered" should be "were altered".

We thank the two reviewers for reviewing our resubmission and providing additional comments. Based on these comments, we have made revisions.

For easy reference, we have color-coded this letter as follows: The reviewers' original comments are in blue. Our responses are in regular typeface. Our changes in response to the reviewers' comments are in red. Finally, our modifications to the original manuscript are underlined in the revised manuscript with track changes.

REVIEWS

Reviewer #1

The authors have done an excellent job in revising their manuscript. I was quite critical regarding the storyline and conclusions when reading the initial version of the paper. Now, I'm pleased to see that the manuscript is much improved and attractive to read. I appreciate that the authors conducted extra experiments. The data on negative-frequency dependent selection has become much stronger and while still negative, we can now reject this hypothesis with high confidence. The evolutionary replay experiment is a great gain for the paper. Although the results do not recover the selective sweeps observed in the initial mono-culture replicates, they clearly show the stabilising effect of *A. johnsonii* in co-cultures. I have a few extra comments.

Thank you very much! We are happy you find the paper improved and the new experiments have strengthened the story. We have addressed your additional comments.

1. The results of the replay experiment could be highlighted more clearly. I propose to mention them in the abstract.

Thank you for this suggestion. **We have highlighted the replay experiment in the abstract.** Please see our comment below.

2. Abstract, lines 28-30. The sentence reads odd as it states that the two morphotypes co-exist (with one of them dominating), while line 24 emphasises a selective sweep. This causes confusion. I assume this sentence refers to the replay experiment. I thus recommend to explicitly state this and to emphasise the key results: *A. johnsonii* has a stabilising effect.

Thank you for pointing out this oddity. You are correct that this sentence referred to the replay experiment, but we see how it came out confusing. Based on your suggestion, we have changed the sentence to explicitly refer to the replay experiment and the stabilizing effect of *A. johnsonii*. **The revised sentence in the abstract (lines 28-30) reads: "Finally, we conducted an**

evolutionary 'replay' experiment to assess whether the presence or absence of A. johnsonii influenced the coexistence of morphotypes at the population level. Interestingly, A. johnsonii had a stabilizing effect on the co-culture."

3. Line 191. In this section, phenotypic assays are described and not physiological tradeoffs. Please clarify.

Thank you for catching our typo. These are indeed physiological assays and not physiological tradeoffs. **We have corrected the heading.**

4. Fig. 6 and lines 399-401. Since there is no frequency effect, it would be possible to pool the data across frequencies to conduct a global analysis with higher statistical power. I.e., one could test whether the relative fitness of the small genotype is lower in co-culture than in mono-culture, which would point towards a stabilising effect in co-culture.

We appreciate this suggestion. We pooled the data across frequencies and conducted a two-tailed t-test to assess whether the relative fitness of the small genotypes was different in co-culture than mono-culture. The differences were not statistically different despite the higher statistical power (two-tailed t-test, d.f. 28, $p = 0.8139$). We also tested whether the relative fitness of the small genotype is lower in co-culture than in mono-culture, as you suggested. Still, the differences were not statistically significant (one-tailed t-test, d.f. 28, $p = 0.407$).

We revised the text to add the results from the two-tailed t-test (lines 398-404): "Finally, we tested whether the relative fitness of the small genotype differed in monoculture than in coculture by pooling data across frequencies. We did not find significant differences in the small genotype's relative fitness, whether A. johnsonii was present or absent (two-tailed t-test, d.f. 28, p-value = 0.814). In conclusion, these results suggest that frequency-dependent selection is not a process driving coexistence between the fleQ and small genotypes from replicate #2, and the presence of A. johnsonii does not influence the relative fitness of the small genotype."

Reviewer #2 (Comments for the Author):

The authors have addressed my concerns - thank you.

Thank you very much!

I have only one minor edit to point out - Line 698 "got altered" should be "were altered".

Thank you for catching our typo. **We have corrected the sentence in the figure legend of Fig.7.**

Re: mSystems01053-24R1 (Microbial diversification is maintained in an experimentally evolved synthetic community)

Dear Dr. Alejandra Rodríguez-Verdugo:

Your manuscript has been accepted, and I am forwarding it to the ASM production staff for publication. Your paper will first be checked to make sure all elements meet the technical requirements. ASM staff will contact you if anything needs to be revised before copyediting and production can begin. Otherwise, you will be notified when your proofs are ready to be viewed.

Sincerely,
William Harcombe
Editor
mSystems